# Adolescents’ Online Pornography Exposure and Its Relationship to Sociodemographic and Psychopathological Correlates: A Cross-Sectional Study in Six European Countries

**DOI:** 10.3390/children8100925

**Published:** 2021-10-16

**Authors:** Elisabeth K. Andrie, Irene Ikbale Sakou, Eleni C. Tzavela, Clive Richardson, Artemis K. Tsitsika

**Affiliations:** 1Adolescent Health Unit, Second Department of Pediatrics, P. & A. Kyriakou Children’s Hospital, School of Medicine, National and Kapodistrian University of Athens, 11527 Athens, Greece; isakuped@yahoo.gr (I.I.S.); et@intec.gr (E.C.T.); info@youth-health.gr (A.K.T.); 2Department of Economic and Regional Development, Panteion University of Social and Political Sciences, 17671 Athens, Greece; crichard@panteion.gr

**Keywords:** adolescence, Europe, online, pornography exposure, prevalence, psychosocial functioning

## Abstract

The aim of the study was to assess the prevalence of online exposure to pornography in European adolescents and its relationship to sociodemographic and psychopathological correlates. A cross-sectional school-based survey of 10,930 adolescents (5211 males/5719 females), aged 14–17 years old (mean age 15.8 ± 0.7) was carried out in six European countries (Greece, Spain, Poland, Romania, the Netherlands, and Iceland). Anonymous self-completed questionnaires covered exposure to pornography, internet use and dysfunctional internet behavior, and psychopathological syndromes (measured by Achenbach’s Youth Self-Report). The prevalence of any online exposure to pornography was 59% overall and 24% for exposure at least once a week. The likelihood of online exposure to pornography was greater in male adolescents, heavier internet users, and those who displayed dysfunctional internet behavior. Country-specific analyses confirmed that the gender effect existed in every country, although its strength varied, from an odds ratio of 1.88 in Poland to 14.9 in Greece. Online exposure to pornography was shown to be associated with externalizing problem scale scores, especially rule-breaking and aggressive behavior, but also associated with higher scores in competences, namely activities and social competence. Exposure to pornography is ubiquitous, more relevant to boys, and is associated with both positive qualities/competences and externalizing behavioral problems.

## 1. Introduction

The great increase in internet usage has brought with it ubiquitous access to online pornography, defined as watching online pornography or accessing sexually explicit content on the internet. Triggered by biological, social, and cognitive changes, adolescents tend to be more susceptible to exposure to pornography [1,2,3,4]. The normative development of sexuality that reaches its peak during adolescence, sexual curiosity and an increasing need for sexual information [1,2,3], and ineffective parent–child communication related to sexuality and limited formal sex education [4,5] all contribute to adolescents’ increased exposure to pornography. Lack of sex education may be especially relevant in some counties due to socio-cultural norms. Furthermore, the easy accessibility, anonymity, and affordability, described by Cooper [6] as the Triple A Engine about twenty years ago, are still pertinent factors and facilitate the phenomenon of increased online exposure to pornography. Additionally, Griffiths [7] suggested factors such as: convenience, escape, and social acceptability. 

Adolescents’ exposure to pornography has raised concerns in light of potential risks to adolescent development. In particular, the impact of pornography on adolescents’ cognitive, behavioral, and emotional responses has been widely investigated and reviewed [8,9]. Apprehension remains prominent about the negative impact of pornography on youth [8,10,11,12,13,14], yet research on the subject is inconclusive, and even positive effects of pornography have been reported [15,16]. Behavioral science research has shown that exposure to pornography can influence youth sexual attitudes, which, in turn, impact their sexual behavior [8,17] and socio-emotional functioning [12,18,19,20]. This may be especially relevant when exposure is frequent and may affect some adolescents more than others. As previously noted, it is necessary to embrace a developmental and socio-cultural perspective in assessing the impact of adolescents’ use of pornography [21].

### 1.1. Divergent Prevalence of Adolescents’ Use of Pornography

Many studies related to adolescents’ use of pornography have shown that prevalence rates of exposure vary greatly across studies and countries. Some very high prevalence rates up to 98% have been reported, for example, in Germany, Sweden, Poland, and Italy [8,22,23,24,25]. Moderate rates, from 36% to 57%, were noted among other European countries, such as Belgium, Switzerland, the Netherlands, Greece, and the Czech Republic [26,27,28,29,30]. The U.S. is also characterized by a wide variation in the prevalence of pornography exposure, with marked fluctuations noted in the last 20 years [31,32].

The different methodologies used (i.e., self-administered versus interviewer-administered) [33], especially in adolescents [34], may account for different prevalence figures [35]. Additionally, inconsistencies in operational definitions of pornography use throughout the literature [9] may be yet another factor responsible for differences in prevalence estimates. Furthermore, many studies assess occasional use, which may be unintentional, while others focus on regular use, which is typically intentional. 

### 1.2. Factors Associated with Adolescents’ Pornography Exposure

Adolescents’ exposure to pornography is influenced by a range of developmental, social, demographic, psychological, and educational factors. Most studies have shown that male adolescents used pornography more often than their female counterparts [3,21,32,33,36,37,38,39,40,41,42]. However, gender differences in exposure to pornography were not shown in more liberal countries [38,43,44,45]. Regarding age, discrepancies have emerged. Some studies have shown that older adolescents are much more likely to have been exposed to pornography than younger adolescents [32,38,41,42], but others do not support this association [21,45]. In a European study by Stanley et al., young adolescents reported less interest in watching pornography after they became sexually active [40]. Other sociodemographic measures that have been found to be associated with more frequent exposure to pornography include poor family functioning in terms of less mutuality, communication and harmony [5,46,47,48], low socioeconomic status [1], and less educated parents [4,49,50].

Adolescent Internet pornography viewing has increased significantly in the last decade, with research highlighting its association with Internet addictive behavior (IAB) [51,52]. IAB, defined as loss of control over internet use, and risk of IAB (fewer or weaker symptoms of IAB), collectively referred to as Dysfunctional internet behavior (DIB) [51], represents a real threat to adolescents. In the EU NET ADB project, specific online activities such as gambling, social networking, gaming, and internet pornography use were shown to be associated with greater probability of reporting DIB [51]. There is currently a gap in the knowledge base of the relationship between exposure to internet pornography and adolescents’ patterns of internet behavior, which the present study aims to fill. Ševčíková and Daneback [32] suggested that the use of online pornography was higher among adolescents who use the internet more often. Visiting pornographic sites has also been identified as a risk factor for excessive or pathological internet use among Greek adolescents [52], [33]. However, pathological internet use or DIB is not restricted only to the extent and frequency of internet use but is a complex entity that also includes a series of behaviors related to internet use that may lead to impairment or distress [53,54,55]. 

Based on the literature to date, it is hypothesized that:(1)adolescents who show patterns of excessive internet use will present an increased likelihood of consuming pornographic material via the internet;(2)adolescents who report exposure to pornography will present higher levels of behavioral and emotional problems.

The relationship of exposure to online pornography to adolescent behavioral and emotional problems has not been extensively studied previously in Europe. To our knowledge, ours is the first study that has implemented an in-depth psychosocial functioning assessment in relation to exposure to online pornography in adolescence, which we hypothesized to be correlated with it. Such exploration can enrich our understanding of the factors that are correlated with exposure to pornography. If further investigation confirms causal relationships, this can feed into prevention and intervention programs and can inform the design of sexuality education.

The present study undertook the examination of exposure to online pornography in a large representative sample of 14–17 year-old adolescents who completed a survey with common procedures and a common questionnaire in six countries. This enables a valid comparison of prevalence rates and relevant risk factors, which we hypothesized to be correlated with it. The specific objectives of the present study were: (1)to estimate the prevalence of exposure to online pornography among 14–17 year-old adolescents across six European countries;(2)to examine the association between adolescents’ internet exposure to pornography and a number of sociodemographic and internet use factors;(3)to investigate the association between exposure to pornography and measures of psychosocial functioning–psychopathological syndromes (behavioral and emotional problems).

## 2. Methods

### 2.1. Participating Countries and Procedure

A cross-sectional study was conducted as part of a larger mixed-methods research project on the intensity and prevalence of internet addictive behavior risk among minors in Europe across seven countries: Greece, Spain, Poland, Germany, the Netherlands, Romania, and Iceland. Germany was excluded from the present analysis because its questionnaire did not include the assessment of psychosocial functioning. The study protocol was approved by the appropriate ethical committees of participating countries. Each country drew a school-based clustered probability sample from official national lists, with school class (ninth and tenth grade) as the primary sampling unit. About 100 classes (2000 students) were sampled in each country. Data were collected from October 2011 to May 2012. The sampling and data collection procedures have been described in detail previously [51]. All students attending the selected classes who were present on the day of data collection were eligible to participate if they produced a signed parental consent form that had been distributed to their legal guardians prior to the execution of the study, emphasizing the confidentiality and anonymity of the study. Anonymous pencil-and-paper questionnaires were completed individually by participants in the classroom during one school hour under the supervision of a trained research assistant without the teacher’s presence. In total, approximately 10% of registered students were absent on the day of data collection, and 3% of those present either refused to participate or did not have the necessary permission. Of the 11,298 adolescents who completed the survey, 100 fell outside the eligible age range of 14–17.9 years. An additional 268 participants (2.4%) were excluded due to unknown age or gender. This left a final sample of 10,930 adolescents.

### 2.2. Measures

#### 2.2.1. Sociodemographic Variables 

Year and month of birth and gender were reported. Educational attainment of parents was used as a proxy mixture of socioeconomic status. Specifically, educational attainment was measured by the highest qualification earned between the two parents. Two categories were created: low/middle (primary or secondary school) and high (postsecondary or tertiary education) educational level.

#### 2.2.2. Exposure to Pornography 

Adolescents were asked whether they had seen pictures or videos with sexually explicit content on the internet in the last 12 months. The question was preceded by an introduction providing examples of what is to be understood as sexually explicit online content (“naked people or people engaged in sexual intercourse”). Response options were yes/no/prefer not to say. Those who answered yes were then asked to report on the frequency of this behavior (“how often did you see such content online in the last 12 months”) with response options of every day or nearly every day/once or twice a week/once or twice a month/less often.

#### 2.2.3. Intensity of Social Networking Sites (SNS) Use 

Adolescents were asked for how long they had used SNS on a typical weekday (“normal school day”) and at weekends or during vacation (“non-school day”) during the past 12 months. Response options ranged from “not at all” to “more than four hours”. The weighted average of weekday and weekend use provided a single estimate of daily SNS use. The median response “2 h per day” was used to dichotomize the amount of time spent on SNS into moderate use (<2 h daily) and heavier use (≥2 h daily). 

#### 2.2.4. Internet Addictive Behavior 

The Internet Addiction Test (IAT; Young, 1998) was administered. This 20 item scale—which was already in use in each participating country—evaluates the degree of preoccupation, compulsive use, behavioral problems, emotional changes, and impact of internet use upon functioning. Following a pilot study within this project, the phrasing of three items was modified for adolescents and contemporary internet use. For example, item 3, “how often do you prefer the excitement of the internet to intimacy with your partner?” was modified to “how often do you prefer the excitement of the internet to being with your boyfriend or girlfriend?” Response scores to each item range from 0 to 5, where a score of 0 corresponds to responses of “never/not applicable”, 1 corresponds to ‘‘rarely’’ and 5 to ‘‘always’’. The total IAT score thus ranges from 0 to 100. In order to assess internet addictive behavior, the following cutoff scores were recommended by Young: (a) 0–19, no signs of internet addictive behavior; (b) 20–39, mild, yet non-problematic signs of internet addictive behavior; (c) 40–69, “at risk for internet addictive behavior”; (d) 70–100, internet addictive behavior. We grouped the first two categories together as functional internet behavior (0–39 points) and the other two together as dysfunctional internet behavior (DIB; 40–100 points). Up to two missing values per participant were permitted and were replaced by the country-specific median. The test showed very high internal consistency in each country in this study (Cronbach’s alpha from 0.907 to 0.925).

#### 2.2.5. Emotional and Behavior Problems 

We used the Youth Self Report (YSR) Problem Checklist to measure emotional and behavior problems [56], an empirically derived and widely used 112 item self-report instrument for adolescents 11–18 years of age with excellent psychometric properties. The YSR is used in both clinical and school settings in assessing adolescents’ behavioral and emotional problems as well as their competences. Adolescents rate how true each item is for themselves (now or within the past 6 months) using a 3 point scale (0 = absent, 1 = occurs sometimes, 2 = occurs often). The YSR generates a total of 25 scores of competences and problem behaviors. The YSR was available in a translated and standardized version in all participating countries. Our analyses focused on syndrome scales and the broadband clusters of internalizing and externalizing problems. An example item from the anxious/depressed syndrome scale is: “I am self-conscious or easily embarrassed.” Adolescents rate how true each item is of themselves now or within the past 6 months, using a 3 point scale (0 = not true, 1 = somewhat/sometimes true, 2 = very true/often true). Raw scores were used in our analyses, with higher scores indicating more problems. Cronbach’s alpha coefficients were calculated separately in each country; the lowest values were 0.78 for anxious/depressed, 0.70 for withdrawn/depressed, 0.75 for somatic complaints and 0.77 for the clustered internalizing problems. 

#### 2.2.6. Competences (Offline) 

We used the YSR activities, social competence, and academic performance scales. Activities capture the sports and hobbies that adolescents participate in, and rate time and competence in them. Social competence assesses the number of close friends and the frequency and quality of social interactions. (For example, “Compared to others of your age, how well do you get along with other kids?”). Raw scores were used in analysis, with higher scores indicating greater competency. Cronbach’s alpha coefficients, calculated separately in each country, were acceptable for activities (smallest in any country was 0.56) and for academic performance (smallest 0.64). Values for social competence were low, ranging from 0.24 to 0.44 across countries. However, for completeness, it was included in the analysis with the rest of the YSR scales. 

### 2.3. Statistical Analysis 

For the comparison of proportions, Pearson’s chi-squared test of independence was used. Multiple logistic regression analysis was employed to identify factors independently associated with exposure to pornography and, separately, for frequent exposure. This was carried out for the total sample and for each country separately. The independent variables entered into the model were country (in the overall analysis), gender, age, highest educational level of the parents, presence of DIB, hours of use of the internet and of social networking sites, and hours spent computer gaming. The overall analysis was also performed, including interaction terms between country and the other independent variables, in order to identify which factors had different effects between countries. Adjusted odds ratios are presented for the results of the logistic regression analyses. 

In order to explore the association between YSR scores and exposure to pornography, linear regression analyses were conducted with dependent variable as each YSR subscale, and independent variables were the same as those listed above for logistic regression, plus exposure to pornography. For comparability between scales, the effect of exposure to pornography, adjusted for the other independent variables, was expressed as its regression coefficient in standard deviation units of the dependent variable. 

Analyses were conducted using the complex samples procedure in SPSS statistical software (Version 26), so that the computation of all statistical tests and confidence intervals correctly took into account the complex (clustered) sample design.

## 3. Results

A total of 10,930 adolescents’ responses were included in the current study (5211 males/5719 females). Table 1 presents the demographic characteristics of the sample and the prevalence rates of exposure to online pornographic material. 

Overall, as shown in Table 1, 59% had seen pornography in the last 12 months, and 42% of them (23.5% of the total sample) had done so at least once a week. Exposure to pornographic material in the last year varied significantly between countries, ranging from 51.5% in Romania to 67.3% in Poland. Frequent exposure (at least once a week) also varied significantly across countries, from below 20% of the total sample in Romania to 27% in Greece. There was a large gender difference, with 76.5% of males stating that they had seen pornography (40% at least once a week) compared to 42.9% of females (8.9% at least once a week). Differences in parental educational level were much smaller, and age differences were not statistically significant. 

The results of multiple logistic regressions (Table 2) showed independent associations of country, gender, age, parental education, DIB, and the intensity of SNS use, internet use, and gaming with any exposure to pornography in the last 12 months. The adjusted odds ratio (aOR) for male gender versus female was 4.84, with 95% confidence interval (CI) 4.19–5.58. Apart from this, the strongest associations of exposure to pornography were with heavier (>2 h daily) internet use (aOR 1.67, 95% CI 1.44–1.95) and the presence of DIB (aOR 1.59, 95% CI 1.33–1.90). 

In analyses for frequent exposure (Table 3), significant interactions were found between country and gender, country and DIB, country and daily internet use, and, marginally, country and SNS use. In separate analyses for each country, gender had a significant effect in every country, with males everywhere at higher risk than females of having been exposed to online pornography, but the size of this effect was not consistent across counties. A notably strong gender effect was found in Greece and Iceland, where boys were 18 and 7 times, respectively, more likely to be frequently exposed to online pornography, and the weakest gender effects were found in Romania. For intensity of SNS use, the significant interaction effect appears to be due to the quite wide range of odds ratios, from 0.59 in the Netherlands to 1.69 in Greece. The lack of significant interaction with country of age, parental education, and gaming shows that their significant effects on exposure overall do not differ between countries.

With regard to indices of emotional and behavioral problems and competences,Table 4 shows the comparison of mean scores on all YSR scales between adolescents who had been exposed to pornography in the last 12 months and those who had not. The coefficients for activities and social competence were greater among adolescents exposed to online pornography, as compared to their counterparts who reported no such exposure, although the effects were very small (0.10 and 0.06, respectively; Table 4). The regression coefficient for exposure to pornography was not statistically significant for academic performance. All problem scales scores were higher among pornography viewers than non-viewers. The largest effect sizes were for externalizing problems: rule-breaking behavior, aggressive behavior (0.40 and 0.28, respectively). Among internalizing problems scales, attention and thought problems were shown to have the largest effect sizes, yet effects were small (0.25 and 0.22, respectively). 

## 4. Discussion

This study revealed complex patterns in adolescents’ online pornography exposure across six European countries and its associations with a number of individual factors. Overall, more than half the 14–17 year-old adolescents covered by this survey reported that they had been exposed to online pornography during the previous year, and a considerable number (42%) of these were exposed frequently. These figures are higher than those reported in the EU Kids Online study, which was carried out two years earlier in 25 European countries using similar questions. EU Kids Online found that 36% of teenagers aged 15 to 16 years old reported contact with online pornography in the past 12 months [57]. This difference could reflect a rise in rates of exposure over the two-year gap between studies or may be due to methodological differences between the two studies: EU Kids Online was based on face-to-face interviews conducted individually in homes, which may have generated under-reporting, whereas our study was carried out in school classrooms, using anonymous self-completed questionnaires. The latter may have led to higher and probably less biased rates in our study.

### 4.1. Cross-National Patterns of Frequent Exposure 

With regard to frequent exposure, exposure rates found in our study are similar to those of a European school-based survey, conducted in Bulgaria, Cyprus, England, Italy, and Norway at the same time as our study and in the same age group, in which the rates of regular viewing of online pornography varied between 19% and 30% [40]. In our study, frequent exposure (% of those with any exposure), which presents the most meaningful population estimates of accessing pornography [32,33], which we assume to be done intentionally, was reported equally across participating countries (Greece, Spain, Poland, Romania, Iceland) with mild deviation for Netherlands. This similarity suggests that biological factors governing the development of sexuality during adolescence [2] may exert a more powerful impact on pornography than sociocultural ones do. 

Looking at country-specific patterns, in the Netherlands, exposure to pornography has been extensively studied, and the prevalence of purposeful use of pornography in different operationalizations ranged from 28–38% among boys [21]. These rates have remained constant across time, as shown in our study. On the contrary, in Greece, we observed an upward trend in the prevalence rates for any/occasional exposure (57%) and for frequent exposure (49%) compared to those found in a previous study carried out few years ago in the same age groups (19% and 42%, respectively) [29,51]. The availability of formal and probably informal sex education may play an important role in these differences. In Greece, for example, and in contrast with the Netherlands, sex education is not formally included in the school curriculum. This may explain in part the growing exposure rates among Greek adolescents, given the fact that Tsitsika and colleagues have shown that a considerable percent of Greek adolescents utilize internet pornography as a source of sexual information. However, even in other countries that do have formal sex education, educational material may lack images of real bodies and details about sexual practices may be missing, which may inflate the power of pornographic discourse that offers this information [58]. EU-wide school-based sex education programs may provide a preventive measure to minimize over reliance on informal online sex education, and to ensure that adolescents receive valid and safe sex advice and do not (over)rely on online pornographic material for such information.

### 4.2. Gender and Age Effects on Exposure

A strong finding of our study was the consistent existence of significant gender effects across participating countries, replicating a finding of previous research [21,28,29,36,37,38,39,40]. A much higher proportion of adolescent males than females reported exposure to pornography in the last 12 months, and this pattern held true for both any/occasional exposure and frequent exposure. Notably, the size of the gender effect varied across countries, especially in frequent exposure. Males in Greece were 18 times more likely to be involved with pornography than females, while the gender effect was less pronounced among Polish and Romanian adolescents, with boys about four times more likely than girls to be exposed frequently. Why are male adolescents more involved with pornography than girls, and what are the reasons behind differences in pornography exposure between males across countries? We suggest that gender differences could be mediated by divergence in cultural backgrounds, sexual education, or sexual liberalism [21], as well as personality traits, attitudes about sex, and parental monitoring.

Additionally, boys have been previously shown not only to have greater contact with pornography but also to be more likely to be exposed at an earlier age, to see more extreme images [3], and to be more involved in sexting [40,59]. The developmental needs determined by sexual urges, such as getting sexually aroused, and the debut of masturbation that is influenced by androgen hormones [60] include some of the basic reasons for higher pornography uptake among boys. In contrast, girls usually seek pornography in the context of romantic relationships [28] and with curiosity being the most common motive [61]. Additionally, cultural attitudes in terms of promoting masculinity that are observed in some cultures may contribute to this large gender difference. However, further investigation is needed in order to understand this relationship in depth as a contribution to the provision of effective gender-specific prevention programs. 

With regard to age differences, the overall odds of being exposed to pornography were slightly greater for older adolescents, but the age effect was not replicated in country-specific analyses. This small effect may be attributed to the fact that our study did not cover a wide range of ages. However, the previous research findings related to age are inconsistent. Adolescence is characterized by large individual differences in development and biological maturation. Accordingly, more advanced pubertal maturation, especially for boys, has been found to be a more consistent predictor of increased pornography use than chronological age [21]. 

In contrast with previous studies [4], higher parental education level was associated with adolescents’ frequent exposure to pornography in our study. This effect may demonstrate differences in parental attitudes on pornography exposure, with more educated parents possibly being more open-minded towards pornography [62]. Additionally, since parental education was used as a proxy for socio-economic level, this difference may be explained in terms of increased availability of appliances and internet access or increased privacy due to having one’s own room, which has been previously shown to increase use [57].

### 4.3. Internet Use: DIB but Not SNS Is a Risk Factor for Frequent Exposure 

In terms of adolescents’ internet behavior, we found that adolescents with DIB were twice as likely to have been frequently exposed to internet pornography than those who showed functional internet behavior. This is in line with previous reports in the literature that adolescents who show patterns of excessive internet use presented increased likelihood of consuming pornographic material via the internet [29,45]. Pornography has been shown to be an online application associated with increased risk of developing an addictive internet usage pattern [52,54,63]. In adolescence, the underlying process may be poor parental supervision, limited environmental opportunities for offline experiences, or adolescents’ ascribing excessive importance/value to online activities or reporting underdeveloped self-regulation [64] or underdeveloped behave ioral self-control, which is a marker of internet addictive behaviors [65]. In contrast, the heavy use of social network sites (SNS) was less strongly associated with exposure to pornography. Social networking is not inherently associated with exposure to pornography, because SNS sites do not provide access to such material [66]; the two may in fact be “opposite” activities, with heavier SNS use mostly undertaken by girls and socially oriented adolescents [67]. Hence, an adolescent’s exposure to internet pornography seems to be developed mostly on the grounds of DIB and is facilitated, as has previously been described, by other internet activities such as accessing the internet for sexual education [29], talking to strangers [41], and internet gaming [29].

### 4.4. Are Adolescent Behavior Problems and Competences Associated with Exposure?

The present study advances knowledge of the prevalence of exposure to internet pornography and its emotional and behavioral correlates (competences and problems). Interestingly, adolescents who reported exposure to pornography also received significantly higher scores in Activities and Social Competence scales compared to their peers who reported no exposure, indicating a behavioral pattern characterized by behavioral activation or approach orientation. It may be the case that some adolescents with all-round activity, i.e., offline socially active and participating in sports and other activities, may also be (over)active online, thus also actively seeking out pornographic material. This seeking may be conceptualized within the scope of adaptive adolescent sexual experimentation. The common underlying process between involvement in offline activities and exposure may be behavioral disinhibition [68], which is marked by personality characteristics such as novelty seeking, typically exhibited as exploratory activity in pursuit of rewards and avoidance of monotony [69], increasing the chances of exploratory pornography. Similarly, in a qualitative study of adolescents who were highly engaged online, one of four “digital profiles” was that of the adolescent juggler, active all round and exhibiting an adaptive psychosocial profile, including active engagement in social and extracurricular activities [64]. Along the same lines, in another study, adolescents who were heavily engaged in SNS reported higher social competence scores compared to those with less than two hours of daily SNS use [67]. 

On the negative side of behavioral patterns, a small number of previous studies have addressed the impact of exposure to pornography on adolescent behavioral problems: rule-breaking or highly delinquent youth [41,70] and youth showing symptoms of borderline or clinical depression have been reported to use pornography more frequently [41,71,72]. Similarly, Ybarra and Mitchel [42] (2005) linked pornographic consumption with behavioral problems and depressive symptoms. Accessing online pornography may be a dysfunctional way to cope with stress or with abnormal mood, and as such, it is reinforced and maintained [73]. Accordingly, Brand, Laier, and Young [74] suggested that problematic internet use is associated with expectations that the internet can positively influence mood, the disappointment of which may in turn worsen preexisting mental health problems. Alternatively, the content of the pornographic material may impact negatively on youth by shaping adolescent beliefs and their attitudes towards sex, which may, in fact, come into conflict with those instilled by their families [75]. Tsitsika et al. suggested that Greek adolescents might develop unrealistic attitudes towards sex through exposure to online pornography. These conflicts need to be addressed in sex education classes. Psychosocial impairment associated with pornographic exposure suggests a need to further explore and address underlying mechanisms reinforcing pornographic use behavior among some adolescents.

Regarding aggressive behavior, Mesch, in his study of the social characteristics of pornography users in a sample of Israeli adolescents aged 13–18 years, found a significant association between pornography consumption and aggressiveness in school, with higher degrees of consumption related to higher levels of aggressiveness [45]. In a similar study discussed previously, Alexy, Burgess, and Prentky [76] found that juvenile sexual offenders who were consumers of pornography were more likely to display forms of aggressive behaviors such as theft, truancy, manipulation of others, arson, and forced sexual intercourse [76]. The effect of androgens in both sexual and aggressive behavior shown in the past could be a possible mediator of these relationships [18]. Testosterone levels are positively correlated with the level of aggressiveness, violent behavior, and sexual offences, particularly among adolescents [77,78]. However, well-designed studies are necessary in the future in order to clarify the causality of this relationship.

## 5. Strengths and Limitations

A major strength of this study is its large sample of adolescents obtained from probability samples in six European countries. Students were surveyed using common questionnaires, with the same translation and data collection procedures in each country, thus ensuring the validity and comparability of the data. The present study is the first of its kind to assess the emotional and behavioral characteristics of adolescents using an empirically derived and widely applied self-report instrument validated in each country. 

Some limitations of the study should be acknowledged. First, data are cross-sectional, and so do not provide information about the causal relationship between pornography exposure and mental health problems. Longitudinal research is warranted to further explore pornography exposure and its mental health implications. Second, data are self-reported and, as such, are subject to social desirability, to an extent that may vary between countries. Thirdly, exposure to pornography was not specified as being accidental or sought after, although frequent exposure is intuitively not accidental. Fourthly, the scale of social competence had low reliability, and results based on this are more tentative than others. Moreover, this study was conducted in 2011–2012 across six European countries, thus limiting the extent to which it represents the current picture of adolescent’s pornography exposure, bearing in mind the rapid technological developments in Internet, and especially mobile, technologies. Nevertheless, no more recent research on nationally representative samples of European adolescents has yet been published. Therefore, further studies should evaluate whether the associations persist in the 2020s. Finally, six European countries participated in the study; our findings are not necessarily generalizable to the rest of Europe or to other parts of the world.

## 6. Conclusions

This study showed that exposure to pornography is normative for today’s adolescents, with one in four European adolescents frequently exposed to online pornographic material and relatively little variation in rates across participating countries, suggesting underlying biological mechanisms. Boys were shown to be at higher risk than females of having been exposed to online pornography, but the size of this effect was more pronounced in some countries than others, suggesting cultural and parental mediation effects. We found that adolescents exhibiting DIB had greater odds of exposure to internet pornography than those showing functional internet behavior, which may suggest an overlap between the two clinical entities or of common underlying behavioral self-control. Importantly, exposure to pornography was shown to be associated with externalizing problem scales, especially rule-breaking and aggressive behavior, but also associated with higher scores in competences, namely activities and social competence. 

## Figures and Tables

**Table 1 children-08-00925-t001:** Any and frequent (at least once a week) exposure to pornography on the internet in the last 12 months (% based on valid responses) *.

		Any Exposure to Pornography		Frequent Exposure		
	Total Sample	n	%	n	% of Total Sample	% of Those with Any Exposure
Total	10,930	5736	58.8	2141	23.5	42.0
Country						
Poland	1978	1182	67.3	444	26.8	41.0
Iceland	1926	1138	65.0	381	24.1	39.3
Greece	1967	1051	57.2	471	27.0	49.0
Netherlands	1249	626	57.0	202	19.7	36.7
Spain	1980	966	53.5	376	21.9	43.0
Romania	1830	773	51.5	267	19.2	40.2
Gender						
Male	5211	3534	76.5	1712	40.0	53.6
Female	5719	2202	42.9	429	8.9	22.5
Age						
14–15 years	6805	3603	58.9	1347	23.6	42.1
16–17 years	4125	2133	58.7	794	23.3	41.7
Parental education						
Low/middle	3590	1819	56.1	641	21.1	39.7
High	5688	3164	61.4	1223	25.3	43.0
Daily SNS use						
≤2 h	6631	3334	55.9	1203	21.4	40.4
>2 h	3648	2139	65.3	855	28.0	44.6
Dysfunctional internet behavior						
No	9002	4625	57.5	1601	21.3	39.0
Yes	1537	988	71.4	490	37.7	54.2
Daily internet use						
≤2 h	4963	2253	50.3	704	16.7	31.2
>2 h	5162	3140	67.5	1310	30.1	41.7
Daily gaming						
≤2 h	6595	2980	50.6	848	15.3	28.5
2 h	3381	2179	72.0	1035	36.7	47.5

* All differences between groups are statistically significant (*p* < 0.001 in chi-squared tests) except for differences between age groups.

**Table 2 children-08-00925-t002:** Results of logistic regression analyses for any use of pornography in the last 12 months. Table shows odds ratios (adjusted for the other explanatory variables included in the analysis) with 95% confidence intervals.

	Overall *	By Country					
		Greece	Netherlands	Romania	Poland	Iceland	Spain
Gender							
Male vs. Female	4.84 (4.19–5.58)	14.9 (11.2–19.8)	4.69 (2.96–7.44)	4.06 (2.91–5.65)	1.88 (1.35–2.63)	2.39 (1.54–3.71)	5.71 (4.47–7.28)
Age							
16–17 vs. 14–15	1.25 (1.10–1.42)	1.29 (0.94–1.77)	1.59 (1.07–2.35)	1.58 (1.12–2.23)	0.93 (0.69–1.26)	1.34 (0.92–1.96)	1.16 (0.90–1.51)
Parental education							
High vs. Low/Middle	1.20 (1.07–1.35)	1.15 (0.84–1.57)	1.55 (0.99–2.42)	1.22 (0.95–1.57)	1.09 (0.82–1.45)	1.07 (0.83–1.37)	1.25 (0.97–1.61)
Internet Behavior							
Dysfunctional vs. Functional	1.59 (1.33–1.90)	1.39 (0.86–2.23)	2.42 (1.36–4.31)	1.56 (1.07–2.28)	2.19 (1.23–3.91)	1.29 (0.74–2.25)	1.65 (1.19–2.31)
Daily use of Social Networks							
>2 h vs. ≤2 h	1.19 (1.02–1.39)	2.10 (1.39–3.18)	0.76 (0.44–1.34)	0.97 (0.69–1.37)	1.12 (0.78–1.61)	1.39 (1.03–1.89)	0.98 (0.66–1.46)
Daily gaming							
>2 h vs. ≤2 h	1.18 (1.03–1.35)	1.18 (0.88–1.59)	1.49 (0.94–2.38)	1.14 (0.83–1.57)	0.93 (0.67–1.30)	1.52 (1.08–2.13)	1.91 (1.33–2.74)
Daily internet use							
>2 h vs. ≤2 h	1.67 (1.44–1.95)	1.25 (0.88–1.79)	1.77 (0.96–3.27)	1.83 (1.27–2.62)	2.52 (1.83–3.47)	1.47 (1.05–2.06)	1.54 (1.07–2.20)

* Country also included among the explanatory variables.

**Table 3 children-08-00925-t003:** Results of logistic regression analyses for frequent use of pornography in the last 12 months. Table shows odds ratios (adjusted for the other explanatory variables included in the analysis) with 95% confidence intervals.

	Overall *	By Country					
		Greece	Netherlands	Romania	Poland	Iceland	Spain
Gender							
Male vs. Female	6.80 (5.71–8.10)	18.5 (12.0–28.5)	6.46 (3.13–13.3)	3.50 (2.23–5.29)	3.73 (2.56–5.43)	7.27 (4.82–12.3)	7.94 (5.29–11.9)
Age							
16–17 vs. 14–15	1.15 (0.98–1.34)	0.93 (0.66–1.32)	0.86 (0.51–1.46)	1.39 (0.93–2.08)	1.17 (0.83–1.66)	1.73 (1.14–2.65)	1.13 (0.80–1.60)
Parental education							
High vs. Low/Middle	1.31 (1.14–1.51)	1.42 (1.04–1.95)	1.01 (0.55–1.85)	0.96 (0.72–1.28)	1.64 (1.20–2.24)	1.16 (0.78–1.72)	1.45 (1.07–1.96)
Internet Behavior							
Dysfunctional vs. Functional	1.97 (1.65–2.37)	1.38 (0.84–2.28)	4.11 (2.49–6.78)	1.71 (1.12–2.61)	2.31 (1.54–3.47)	2.82 (1.56–5.09)	2.01 (1.40–2.89)
Daily use of Social Networks							
>2 h vs. ≤2 h	1.21 (1.03–1.43)	1.69 (1.17–2.24)	0.59 (0.32–1.08)	1.00 (0.70–1.42)	1.38 (0.90–2.13)	1.48 (1.02–2.15)	0.77 (0.51–1.18)
Daily gaming							
>2 h vs. ≤2 h	1.27 (1.10–1.47)	1.64 (1.19–2.26)	1.93 (1.08–3.42)	0.97 (0.70–1.35)	1.10 (0.79–1.54)	1.27 (0.87–1.84)	1.64 (1.17–2.29)
Daily internet use							
>2 h vs. ≤2 h	1.68 (1.42–1.99)	1.80 (1.22–2.66)	1.83 (1.08–3.07)	1.74 (1.19–2.55)	1.61 (1.09–2.38)	1.75 (1.11–2.76)	1.87 (1.22–2.84)

* Country also included among the explanatory variables.

**Table 4 children-08-00925-t004:** Comparison of Youth Self-report scale scores between adolescents who had and had not viewed internet pornography in the last 12 months.

Youth Self-Report Scales	Range	Non-Viewers (*M*, SD)	Viewers (*M*, SD)	Effect Size *	*p*
Activities	0–12	8.13 (3.81)	8.18 (3.72)	0.10	<0.001
Social competence	0–12	7.55 (2.43)	7.93 (2.65)	0.06	0.007
Academic performance	0–4	2.20 (0.58)	2.15 (0.61)	0.03	0.18
Total competence	0–28	17.90 (5.54)	18.31 (4.52)	0.09	<0.001
Anxious-depressed	0–26	4.96 (4.29)	5.04 (4.52)	0.12	0.009
Withdrawn-depressed	0–16	3.02 (2.67)	3.14 (2.89)	0.07	<0.001
Somatic complaints	0–20	2.82 (3.11)	3.00 (3.17)	0.15	<0.001
Social problems	0–22	2.89 (2.64)	3.17 (2.92)	0.10	<0.001
Thought problems	0–24	2.85 (3.03)	3.61 (3.54)	0.22	<0.001
Attention problems	0–18	4.76 (3.23)	5.68 (3.40)	0.25	<0.001
Rule-breaking behavior	0–30	3.43 (3.32)	5.43 (4.39)	0.40	<0.001
Aggressive behavior	0–34	6.10 (4.77)	7.75 (5.79)	0.28	<0.001
Total problems	0–230	32.91 (21.34)	38.83 (24.67)	0.29	<0.001

* Coefficient of pornography viewing (yes/no) in multiple linear regression adjusted for country, gender, age, parental education level, presence of DIB, hours of internet use, hours of SNS house, and hours gaming. Coefficient standardized by dividing by SD of dependent variable for comparability.

## Data Availability

The data presented in this study are available on request from the corresponding author.

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
