# Peer review of "Adolescents’ Online Pornography Exposure and Its Relationship to Sociodemographic and Psychopathological Correlates: A Cross-Sectional Study in Six European Countries"

_children, 2021, doi:10.3390/children8100925_

Round 1

Reviewer 1 Report

Thank you for the opportunity to review this paper. It is well-written, concise and makes an important contribution to the literature.

I have only minor comments:

Abstract:

  •  Could the authors specify the gender effects?

Introduction:

  • line 33/34:  "Triggered by biological, social and cognitive changes, adolescents 33 tend to be more susceptible to exposure to pornography."  Could the authors provide a reference?
  • line 56 ff. on prevalences: I find it bit problematic to state prevalence rates, which obviously differ and even point to 98% without trying to explain the big range. 
  • the research question is not made clear enough in my opinion, as the paragraph to internet addition is not sufficiently connected to the rest of the introduction. the reader may be confused whether the study "only" analyzes pornographic exposure or also aspects of internet addition. I encourage the authors to revise this paragraph and to present the research question more clearly. 
  • Hypotheses are missing
  • line 102: please weaken the aspect of risk factors as studies like yours can only provide information about correlates.

Methods: 

  • line 142 seems to be formatting problem (education attainment of parents)
  • analyses: with respect to this large sample size, the reader wonders why no latent analysis was conducted.

The results and discussion are well-written, conclusions are appropriate.

Author Response

We appreciate the reviewer’s thoughtful reading of our work which undoubtedly improve our text. Detailed responses to each critique are provided below.

Please note that page and line numbers refer to the new marked copy of the manuscript that is attached.

Language has been rechecked throughout the paper by a native English speaker.

Thank you for the opportunity to review this paper. It is well-written, concise and makes an important contribution to the literature.

I have only minor comments:

Abstract:

  •  Could the authors specify the gender effects?

Response:  We have followed this suggestion. Page 1, lines 22-24

Introduction:

  • line 33/34:  "Triggered by biological, social and cognitive changes, adolescents 33 tend to be more susceptible to exposure to pornography."  Could the authors provide a reference?

Response:  We have followed this suggestion. Page 1, line 36

  • line 56 ff. on prevalences: I find it bit problematic to state prevalence rates, which obviously differ and even point to 98% without trying to explain the big range. 

Response: We made the text easier to follow by rearranging the relevant paragraphs (“Divergent Prevalence…..”) so that comments regarding the differences now appear in their logical place, after the prevalence figures. In their previous place, before the prevalences, they were likely to be overlooked. Page 2, lines 58-71

  • the research question is not made clear enough in my opinion, as the paragraph to internet addition is not sufficiently connected to the rest of the introduction. the reader may be confused whether the study "only" analyzes pornographic exposure or also aspects of internet addition. I encourage the authors to revise this paragraph and to present the research question more clearly. 

Response:  We have followed this suggestion. This clarification has been added to the text of the revised manuscript. Page 2, lines 86-92.

  • Hypotheses are missing

Response: According to the reviewer’s suggestion we modified the Introduction section and this clarification has been added to the text of the revised manuscript. Page 3, lines 111, 119

  • line 102: please weaken the aspect of risk factors as studies like yours can only provide information about correlates.

Response:  We agree that that the current study doesn’t demonstrate causation. According to the reviewer’s suggestion we modified the Introduction section and this clarification has been added to the text of the revised manuscript. Page 2, lines 112,119.

Methods: 

  • line 142 seems to be formatting problem (education attainment of parents)

Response: You are absolutely right! We have corrected the problem. Page 4, lines 154, 155.

  • analyses: with respect to this large sample size, the reader wonders why no latent analysis was conducted.

Response. Although we are not sure what particular analysis the reviewer has in mind, it is certainly true that this large set of data is open to further analysis. However, we believe that the analysis that was carried out here represents the necessary first step.

The results and discussion are well-written, conclusions are appropriate.

Reviewer 2 Report

 “Adolescents' online pornography exposure: A cross- sectional study in six European countries”

This is an interesting article with a vast number of participants from six different countries in Europe. Nevertheless, the paper has several flaws, and several changes would improve the overall quality of the manuscript and optimize the chances of publication:

  1. Title does not reflect what was measured in the research. Authors measured exposure to pornography, but also internet use, dysfunctional internet behavior, and psychopathological syndromes. A more inclusive title would bring clarity and pertinence to the article.
  2. Data collection took place one decade ago. Authors must provide evidence that these data are still valid.
  3. Instead of referring readers to reference [56] regarding the sampling and data collection procedures please provide a brief description.
  4. Were YAT and YSR versions validated for all six different countries? What is the Cronbach’s score for each version?
  5. Values for Social Competence were should not be used in the analysis since reliabilities of .24 to .44 are unacceptable.
  6. Authors used version 20 of SPSS. This seems like the article was processed so long ago. Authors should update this. SPSS is currently on version 28.
  7. Table 1. Authors should include chi-square results to determine differences between groups of comparison.
  8. Tables 2-3 are hard to follow. No significance levels are presented.
  9. Authors must discuss the relevance of trying to publish data from a decade ago. Hasn’t anything changed since 2011? If anything, authors should fundament and contextualize this temporal frame as a limitation.

Best wishes.

Author Response

We appreciate the reviewer’s thoughtful reading of our work which undoubtedly improve our text. Detailed responses to each critique are provided below.

Please note that page and line numbers refer to the new marked copy of the manuscript that is attached.

Language has been rechecked throughout the paper by a native English speaker.

Reviewer’s comments

This is an interesting article with a vast number of participants from six different countries in Europe. Nevertheless, the paper has several flaws, and several changes would improve the overall quality of the manuscript and optimize the chances of publication:

  1. Title does not reflect what was measured in the research. Authors measured exposure to pornography, but also internet use, dysfunctional internet behavior, and psychopathological syndromes. A more inclusive title would bring clarity and pertinence to the article.

Response. This is an important remark. Please see the new extended title.

  1. Data collection took place one decade ago. Authors must provide evidence that these data are still valid.

Response: Relevant discussion of this point was also added in the “Strengths and Limitations” section. Page 13, lines 483-488.

  1. Instead of referring readers to reference [56] regarding the sampling and data collection procedures please provide a brief description.

Response: We agree that it is desirable for articles to be self-contained as far as possible. However, we would suggest that lines 140-143 concerning sampling and lines 143-146 concerning data collection give the essential information. We made a small addition. The addition of full details for all six countries seems an unnecessary addition, substantially lengthening the article. 

  1. Were YAT and YSR versions validated for all six different countries? What is the Cronbach’s score for each version?

Response: Yes, all scales had already been validated in each country. This was already noted for the YSR and we have added this information for the IAT (Line 177). Values of Cronbach’s alpha have been added for the YSR (Lines 193, 121, 122); some information was already present for the YSR.

  1. Values for Social Competence were should not be used in the analysis since reliabilities of .24 to .44 are unacceptable.

Response: We agree that the evidence is that there is a question mark against the use of this scale. We have modified the text to show that we included it for completeness along with the rest of the YSR (line 221) and added a comment on it in the Limitations (lines 481, 482).

  1. Authors used version 20 of SPSS. This seems like the article was processed so long ago. Authors should update this. SPSS is currently on version 28.

Response: Our apologies for the misprint. In fact we used version 26. Page 6, lines 240-241

  1. Table 1. Authors should include chi-square results to determine differences between groups of comparison.

Response: Almost all these comparisons are significant with p < 0.001. We have added a footnote to Table 1 to state this.

  1. Tables 2-3 are hard to follow. No significance levels are presented.

Response: We have presented 95% confidence intervals for all the adjusted odds ratios, which also provide information about statistical significance.

  1. Authors must discuss the relevance of trying to publish data from a decade ago. Hasn’t anything changed since 2011? If anything, authors should fundament and contextualize this temporal frame as a limitation.

Response: Relevant discussion was also added in the “Strengths and Limitations” section. Page 13, lines 483-488.

Round 2

Reviewer 2 Report

Thank for implementing the requested changes to your manuscript. I believe that the paper is now more scientifically sound. 

It still think that publishing data from one decade ago raises some issues, but I also think that the nature of the study with such a vast number of participants from 6 European countries brings merit to the contribution.